# Study on the Properties of Vertical Carbon Nanotube Films Grown on Stainless Steel Bipolar Plates

**DOI:** 10.3390/ma12060899

**Published:** 2019-03-18

**Authors:** Congda Lu, Fengye Shi, Jing Jin, Xiang Peng

**Affiliations:** Key Laboratory of E&M, Zhejiang University of Technology, Hangzhou 310014, China; lcd@zjut.edu.cn (C.L.); sfyajmy@163.com (F.S.); yanfeng789@126.com (J.J.)

**Keywords:** stainless steel bipolar plate, carbon nanotubes, corrosion resistance, superhydrophobicity, conductivity

## Abstract

Research on the conductivity and corrosion resistance of stainless steel bipolar plates in a proton exchange membrane fuel cell (PEMFC) is commonly performed in a normal-temperature environment (about 20 °C). However, these fuel cells must function in low-temperature environments (lower than 0 °C) in some conditions, such as in vehicle fuel cells and in portable power supplies that operate during the winter in northern China. Stainless steel bipolar plates have higher requirements in terms of their hydrophobic and anti-icing properties, in addition to needing high conductivity and corrosion resistance. In this study, carbon nanotubes (CNTs) are grown on the surface of 304 stainless steel (304 SS) without a catalyst coating by plasma-enhanced chemical vapor deposition (PECVD), which is a simple and cheap method that allows stainless steel to be used as bipolar plates in low-temperature environments. The Raman spectroscopy and scanning electron microscopy (SEM) results show that the CNTs grown on the surface of 304 SS have different morphologies. The stainless steel samples with different CNT morphologies are tested by hydrophobicity and in situ icing experiments to prove that vertical CNTs can achieve a superhydrophobic state and have good anti-icing properties. The interfacial contact resistance (ICR) of the bare 304 SS and the 304 SS with vertical CNTs is compared by voltammetry, and then the corrosion resistances of both types is compared in a simulated PEMFC environment via a three-electrode system. Consequently, the ICR of the 304 SS with vertical CNTs was lower than the bare 304 SS. The corrosion potential was positive, and the corrosion current density was greatly reduced for the stainless steel with vertical CNTs grown directly on its surface when compared with the bare 304 SS. The experimental results show that vertical CNTs have good application prospects as bipolar plates for PEMFCs in low-temperature environments.

## 1. Introduction

Proton exchange membrane fuel cells (PEMFCs) are widely used in transportation, residential power generation and portable power supplies due to their high reliability, high output power and environmental friendliness [1,2]. The bipolar plate is one of the main components in PEMFC stacks and it accounts for approximately 70% of the total weight and 45% of the stack cost [3,4]. Currently, bipolar plate material mainly includes graphite, metal and composites [5,6,7]. Graphite is most commonly used for its good electrical conductivity, thermal conductivity and corrosion resistance. However, graphite is brittle, of low strength and permeable to gases, which results in a highly difficult and costly processing flow channel on each side. Stainless steel offers many advantages over graphite, including good mechanical performance, low gas permeability, relatively low cost and ease of manufacture. Thus, it is considered as a promising candidate for bipolar plate applications [8,9]. The formation of an insulated oxide layer on the surface of untreated stainless steel significantly increases the interfacial contact resistance (ICR) at the bipolar plate/gas diffusion layer (GDL) interface [10]. Moreover, a stainless steel bipolar plate dissolves metal ions that may contaminate the membrane electrodes because the PEMFC works in weakly acidic environments [11]. To solve these problems and improve the performance of stainless steel bipolar plates, it is necessary to modify the surface of the stainless steel or provide a coating protection. Several coating films, such as metal coatings [12,13,14], metal carbon/nitride coatings [15,16,17], conductive polymer coatings [18,19] and carbon coatings [20,21,22], have been proposed for this purpose.

The corrosion resistance and conductivity of a modified stainless steel bipolar plate in a fuel cell operating at a normal temperature will be greatly improved in the future [23,24]. When a fuel cell operates in a low-temperature environment, the water generated by the cathode catalytic layer enters the flow channel through the gas diffusion layer. The droplets that are not discharged outside the battery with the gas will freeze in the flow channel, causing gas supply channel blockage and fuel cell failure [25,26]. Superhydrophobic surfaces are very suitable for use on stainless steel bipolar plates in low-temperature environments due to their unique properties, such as drag reduction [27,28], anticorrosion [29,30], anti-icing [31,32] and self-cleaning [33,34]. The bipolar plate of the superhydrophobic flow channel, under normal operating temperature conditions, can effectively promote the discharge of excess water in the gas diffusion layer and flow channel to improve cell performance [35,36,37]. Therefore, the hydrophobic properties of stainless steel bipolar plates are of great priority in a low-temperature environment. Regarding modification materials, carbon nanotubes (CNTs) are hydrophobic, conductive [38] and corrosion resistant. A special microstructure can be constructed by CNTs on stainless steel, forming a superhydrophobic bipolar plate surface, which is an ideal modification material for bipolar plates in a low-temperature environment. The formation of a superhydrophobic surface using CNTs can mainly be achieved by the following two methods: (1) A CNT coating to increase the surface roughness [39], and (2) a catalyst coated on a stainless steel surface as a substrate to grow vertical arrays of CNTs modified by low surface energy polymers. Studies have found that vertically aligned CNTs grown on stainless steel can greatly improve hydrophobicity [40]. The use of CNT coatings results in poor adhesion between superhydrophobic surfaces and stainless steel. The method of growing vertical CNTs on the surface of the catalyst is a complicated and relatively costly process, and the low surface energy polymer usually does not have good electrical or thermal conductivity. Vertical CNTs directly grown on the surface of stainless steel have a strong binding force to the substrate, and the hydrophobic properties of CNTs are not affected by high and low alternating temperatures [41,42]. The traditional modification method of a stainless steel bipolar plate can effectively improve conductivity and corrosion resistance, but there have been few studies on the hydrophobic and anti-icing properties at low temperatures. However, the modification method of directly arranging vertically aligned CNTs on the surface of stainless steel can improve conductivity and corrosion resistance and provide better hydrophobicity and anti-icing performance.

In the absence of a catalyst coating, CNTs can be directly grown on the surface of stainless steel mainly by hydrofluoric acid (HF) and plasma etching to convert the stainless steel surface film into nanosized catalyst particles [43]. In this study, methane was used as the carbon precursor. Then, the catalyst particle sizes on the stainless steel surfaces were controlled by the HF and plasma etching conditions. Finally, CNTs were directly deposited by plasma-enhanced chemical vapor deposition (PECVD) on the surface of the pretreated stainless steel, without a catalyst, thereby making the process simpler and cheaper. In addition, we studied the effects of CNTs with different morphologies on their hydrophobicity and anti-icing properties in a low-temperature environment. The results prove that the morphology of the vertical CNTs on the stainless steel surface can improve hydrophobicity and anti-icing performance. Then, we compared the ICR and corrosion resistance of the stainless steel with vertical CNTs and bare 304 stainless steel (304 SS). The results indicate that the application of vertical CNTs on 304 SS bipolar plate fuel cells in a low-temperature environment is feasible.

## 2. Experimental Details

### 2.1. Sample Preparation

The sample used in this experiment was laser-cut 304 SS. The size of the cut 304 SS was 10 × 10 × 1 mm. First, the stainless steel sample was pretreated in a series of ways [43]. The main pretreatment steps were as follows: (1) Sand paper was used to remove the oxidation film on the surface of the stainless steel. (2) Next, acetone and alcohol were used successively for ultrasonic deoiling and decontamination. (3) Then, stainless steel was etched with 25% HF for 200 s. (4) Finally, deionized water was used to clean the stainless steel. The pretreated sample was removed from the alcohol, dried in a nitrogen atmosphere and placed in a quartz boat. Along with the sample, the quartz boat was fed into a PECVD reaction tube, which had a diameter of 122 mm and a length of 2.13 m. The reaction tube was evacuated and purged three times with Ar. The CNT growth steps were as follows: First, the tube reactor was heated to 700 °C in a hydrogen atmosphere. The surface of the steel was etched by hydrogen plasma at this temperature. Then, methane was introduced into the reactor and controlled at a ratio of methane to hydrogen of 6:14. The radio frequency (RF) power was 300 W and growth was maintained for 20 min. After growth was complete, the reactor was cooled to room temperature in a hydrogen atmosphere. Different hydrogen plasma etching times were used to grow CNTs with different morphologies on the sample surface. The samples were labeled a–d. An image of a CNT-coated 304 SS sample is shown in Figure 1. Table 1 shows the specific growth parameters for the CNTs.

### 2.2. Sample Test

Scanning electron microscopy was used to observe the uniformity of the CNT distribution, the orientation of growth and the overall structure. Raman spectroscopy is the most commonly used method for the surface research of carbon materials. The position, strength and shape of the Raman peaks can be used to determine the valence bond structure and vibration form of the carbon material, thereby judging the type of material and the degree of graphitization, which is an effective way to test CNTs. The scanning electron microscope used in this experiment was the Sigma Field Emission Scanning Electron Microscope from ZEISS, Jena, Thuringia, Germany. The microscope was used in the high vacuum mode with an operating voltage of 5 kV. The Raman instrument used in this experiment was the DXR Raman spectrometer from Thermo Fisher Scientific. The excitation wavelength was 532 nm and the test range was 1000–1800 ${\mathrm{cm}}^{-1}$.

When the droplet is stationary on the surface of the sample, the contact angle can determine the hydrophobic properties of the solid material. This study used the SDC-100 contact angle measuring instrument from the SINDIN company with an image magnification of 0.7–4.5× and a continuous zoom. The instrument accuracy was ±0.1°. When measuring the contact angle, the measurement was performed at five different positions and the measurement data was averaged. The effect of a superhydrophobic surface against icing performance can be determined by an “in situ icing” experiment [44]. Here, the water used in the experiment was subcooled deionized water (deionized water that was allowed to stand for 30 min in a 0 °C environment in advance), the water volume used was 10 μL and the test environment temperature was −10 °C, simulating a cold-start for a fuel cell, which is less damaging to the fuel cell. The temperature of the sample and the cryostat were both consistent when the sample was placed in the cryostat for 10 min. The transfer of 10 μL of subcooled deionized water to the surface of the sample was performed with a syringe. When the water droplet was stationary on the surface of the sample, the initial time was recorded as 0 s. The icing of the water droplets was recorded with an electron microscope and imaged at regular intervals until the water droplets were completely frozen. The transparency of the water droplets gradually decreased during the icing process. When the water droplets were completely opaque, they were judged to be completely frozen.

Since stainless steel bipolar plates undergo a transition from a low-temperature environment (−10 °C) to a normal operating temperature (80 °C) during a cold-start, the hydrophobicity of the stainless steel bipolar plate cannot change substantially when alternating between high and low temperatures. Therefore, the hydrophobicity of the sample also required high and low temperature resistance tests. The tests used a high and low temperature damp heat test chamber (model: EW0470) to simulate high to low temperature environments. The sample was placed in a high to low temperature damp heat test chamber, and the temperature change was set from −10 to 80 °C over 5 min to simulate the fuel cell cold-start process. The temperature was raised from −10 to 80 °C and then returned to −10 °C. This process was recorded as one cycle. The entire experiment was cycled 50 times. The contact angle was measured and recorded every 10 cycles.

The interfacial conductivity of the sample was characterized by the contact resistance, and the ICR was measured using Wang’s method [45]. The sample was sandwiched between two pieces of carbon paper (a simulated gas diffusion layer using TGP-H-060 carbon paper with a thickness of 0.2 mm) produced by the Toray Corporation of Japan. The outer layer was supported by a copper plate, which was uniformly pressed during the test to apply pressure to the sample. An Instron LEGEND2345 universal testing machine (Boston, MA, USA) was used to control the pressure change. An external circuit provided a constant voltage of 1 V, and the current produced from the circuit under different pressures was measured by a precision multimeter. Finally, the overall resistance value was calculated by voltammetry. An image of the testing device is shown in Figure 2.

The corrosion performances of the bare 304 SS and treated samples under different conditions were evaluated using potentiodynamic and potentiostatic polarization. The corrosion resistance testing was performed by the traditional three-electrode experiment, with a platinum sheet as a counter electrode, a saturated calomel electrode (SCE) as the reference electrode and the sample as the working electrode, with a working area of 10 × 10 mm. To simulate the working environment of the fuel cell, we used a $0.5\;{\mathrm{MH}}_{2}{\mathrm{SO}}_{4}$ + 2 ppm HF solution as an etching solution. The potential sweep range of the potentiodynamic polarization test was ±0.5 V from the corrosion potential and the scan rate was $20\;\mathrm{mV}\cdot \min^{-1}$. The potentiostatic test was carried out in an etching solution at 70 °C. A −0.1 V (relative to the SCE) potential was applied to the H_2_ purge anode and a 0.6 V (relative to the SCE) potential was applied to the O_2_ purge cathode to simulate PEMFC operating conditions.

The stainless steel bipolar plates, with or without CNTs, were assembled into fuel cells, and then the cells’ performance was characterized by a polarization curve. First, the assembled cells were subjected to activation and initial performance testing and purged with Relative Humidity (RH) 20% of the reaction gas until the RH of the outflow gas reached 30% (both the cathode and anode). Second, the cells were placed in a temperature incubator at −10 °C for 2 h to ensure that the cell temperature would be −10 °C. Next, the cells were taken out of the incubator and fed reactant gas at a rate of $30\;\mathrm{mL}\cdot \min^{-1}$. Then, the cells’ performances were characterized by testing the polarization curves of the cells. Finally, the cells were purged and put into the incubator again for the next cold cycle. Each cell repeated the cold-start process five times.

## 3. Results and Discussions

### 3.1. Sample Surface Topography

Figure 3 shows SEM images of the morphology of the CNTs before and after CNT growth for different etching times. Sample A was not subjected to plasma treatment, and the HF-etched stainless steel surface was divided into large islands and particles of different sizes. The carbon source gas grew sporadic CNTs using the stainless steel particles as a catalyst. The diameter of the CNTs was up to 100 nm and the smallest was 10 nm (Figure 3a). After sample B was plasma treated for 5 min, the surface grain boundaries became more obvious, and the carbon source gas grew a large number of CNTs using these nanosized particles as a catalyst (Figure 3b). However, due to the insufficient density of the nanosized particles, there was a lack of sufficient van der Waals forces between the growing CNTs, which caused them to become entangled (Figure 4a). As the plasma treatment time increased to 10 min, the stainless steel surface particles became more rounded, their size more uniform and their density larger (Figure 3c). It can be seen from Figure 4b that the growth direction of the CNTs could only be perpendicular to the surface of the stainless steel, and the diameters of the CNTs were 30–60 nm. When the plasma treatment time was 20 min, the small particles on the stainless steel surface combined and became larger and more rounded stainless steel particles due to the influence of the high temperature and the plasma. Large-particle stainless steel dissolved a large amount of carbon during the growth of the CNTs. When the temperature was lowered, some carbon precipitated from the surface of the stainless steel to form amorphous carbon instead of CNTs (Figure 3d).

The Raman spectrum of the CNT film on the surface of the sample is shown in Figure 5. The results show that sample had only one peak at $1350\;{\mathrm{cm}}^{-1}$, while samples A, B and C had peaks at 1350 and $1580\;{\mathrm{cm}}^{-1}$, which corresponded to the two main peaks for multiwalled CNTs. As shown in Figure 3, the SEM photographs show that stainless steel sample D had no CNTs on its surface, while samples A, B and C had CNTs on their surfaces. Moreover, samples A, B and C had values for $I_{D}/I_{G}$ of 4.5, 1.6 and 1, respectively, and the relative intensities of the D and G peaks ($I_{D}/I_{G}$) indicated the degree of CNT disorder and the density of defects. Therefore, among the four sets of samples, sample C had the best degree of graphitization and the lowest degree of defects. However, the $I_{D}/I_{G}$ value of sample C was still large, indicating that the CNTs grown on the surface had certain defects. The results indicate that the catalytic effects were influenced by the type and distribution of the catalysts, and that the purity of the grown CNTs decreased with the increase of nonuniformity in the catalysts.

### 3.2. Hydrophobicity and Anti-Icing of the Surface of the Sample

The static contact angle measurement results for the water droplets on the surface of the sample are shown in Figure 6. The plasma treatment changed the size and density of the stainless steel particles on the surface of the sample. The density of the stainless steel particles determined the growth density of the CNTs because there were catalytic elements in the stainless steel particles. The surfaces of the modified samples formed various micro-nanostructures with gaps that could capture air to lift the droplets for better hydrophobicity. Therefore, the contact angle of the modified sample was greatly improved when compared with the bare 304 SS contact angle (83.5°). Moreover, the contact angle of the sample surface increased as the density of the CNTs increased. The stainless steel particle density on the surface of sample C was the largest of all the samples, and consequently, sample C had the largest CNT density and surface contact angle. The measurement of the contact angle of sample C was 150° (Figure 6). The dynamic process of the water droplets in contact with the surface of sample C showed that the water droplets did not adsorb to the surface of the sample when they came into contact, and that even if the water droplets that came into contact with the surface of the sample were squeezed, they could still be separated from the surface of the sample. This indicates that sample C had superhydrophobic properties and low adhesion. When the large-particle stainless steel on the surface of sample D was used as a catalyst, the products deposited were mainly amorphous carbon. Compared with CNTs, amorphous carbon had a surface contact angle that rapidly decreased to 106° due to the lack of a surface microstructure.

The measurement results for the contact angle of the sample surface after 50 high to low temperature alternating experiments are shown in Figure 7. The surface contact angle with the sample did not change significantly after alternating between high and low temperatures, as the CNTs on the surface of the sample did not change after alternating between the high and low temperatures. Moreover, the CNTs grown directly on the stainless steel did not easily separate from the surface, meaning that the sample did not lose its good hydrophobicity.

Figure 8 shows the experimental results for the “in situ icing” test. The water droplets were hemispherical on the stainless steel surface, with the largest contact area corresponding to the shortest icing time (443 s) over all the samples. The icing time (931 s) of the water droplet on the surface of sample A was twice that of the bare stainless steel surface, and the water droplets formed a bulge at the top after completely freezing. The icing time (984 s) of the surface water droplets on sample B was larger than sample A, and the shape change of the water droplets during the icing process was not substantially different from that of sample A. The contact area between the surface of sample C and the water droplets was the smallest of all the samples, and a stable spherical shape was formed. The shape of the water droplets changed during the icing process and formed a peach shape after 1638 s, after which the water droplets became completely opaque ice. The water droplet freezing time (809 s) on the surface of sample D was longer than the freezing time on the surface of the bare stainless steel but was shorter than those of samples A, B and C. These data show that the better the hydrophobic properties of the surface of the sample are, the smaller the contact area is, and the longer the freezing time for the water droplets on the surface of the sample is. From a theoretical analysis of infiltration, we can see that there was a large amount of air between the microstructures of the superhydrophobic surface. This formed a natural barrier that reduced heat loss and made the bottom of the water droplets less likely to freeze. Over time, the low temperature caused the top of the spherical water droplets to begin to freeze and form a bulge. The ice crystals gradually grew downward until they were completely frozen, but the water droplets never spread out on the surface of the sample. Although the superhydrophobic surface did not completely prevent the water droplets from freezing, it could prolong the freezing time of the water droplets on the surface of the sample.

Based on the above-mentioned results, the in situ growth of the vertical CNT film on the surface of the stainless steel bipolar plate was effective at improving hydrophobicity and anti-icing performance in a low-temperature environment, and the hydrophobic properties were not affected afterwards by alternating between high and low temperatures. Therefore, the conductivity and corrosion resistance of the modified sample were studied in a follow-up study using sample C as the research object.

### 3.3. Analysis of the Contact Resistance of the Samples

Figure 9 shows the contact resistance of sample C and the bare 304 SS as a function of pressure. As the interface pressure increased, the contact resistance of sample C and the bare 304 SS significantly reduced. Since the interface pressure was small, there was a small contact point between the sample and the carbon paper, resulting in a large contact resistance. Since the contact between the sample and the carbon paper was a point contact when the interface pressure was small, the contact resistance was large. The carbon paper was deformed as the interface pressure increased, and the contact area between the sample and the carbon paper gradually increased, meaning that the interface contact resistance gradually reduced. When the interface pressure increased further, the deformation of the carbon paper reached a limit, and the contact area did not change further, meaning that the contact resistance also remained stable. When the pressure of sample D was greater than $50\;N\cdot {\mathrm{cm}}^{-1}$, the contact resistance was basically stable. Nevertheless, when the pressure of the bare 304 SS was greater than $120\;N\cdot {\mathrm{cm}}^{-1}$, the contact resistance remained stable. The reason for this phenomenon is that the growth of the CNTs on the surface of sample C only required a small amount of pressure for complete adherence to the carbon paper, whereas the pressure required for the bare 304 SS to adhere to the carbon paper was large. As seen in Figure 9, the contact resistance of the bare 304 SS was always greater than that of sample C. The main reason for this is that the conductivity of the oxide film on the surface of the bare 304 SS was worse than that of the CNT film. The assembly pressure of the PEMFC stack was $150\;N\cdot {\mathrm{cm}}^{-2}$, and the contact resistance should be less than $20\;m\Omega \cdot {\mathrm{cm}}^{-2}$ [46]. Under this pressure, the contact resistance of the bare 304 SS was $135.6\;m\Omega \cdot {\mathrm{cm}}^{-2}$. The contact resistance of sample C was $16.3\;m\Omega \cdot {\mathrm{cm}}^{-2}$, which is only 10% of the bare 304 SS’s contact resistance. Therefore, the growth of vertical CNTs on the surface of the stainless steel could effectively reduce the contact resistance and meet the requirements of a bipolar plate in terms of conductivity.

### 3.4. Analysis of the Corrosion Resistance Performance of Samples

Figure 10 shows the potentiodynamic polarization curves of sample C and the bare 304 SS in a simulated PEMFC environment. The corrosion potential of sample D was 0.12 V, which was 0.35 V higher than that of the bare 304 SS. The corrosion current density of the bare 304 SS was $17.5\;\mu A\cdot {\mathrm{cm}}^{-2}$, and that of sample C was $1.94\;\mu A\cdot {\mathrm{cm}}^{-2}$. The corrosion current density of sample C was one order of magnitude lower than that of the bare 304 SS. This is mainly because the vertical CNTs on the surface of sample C could effectively isolate the stainless steel substrate and etching solution. The more positive the corrosion potential is, the smaller the corrosion current is, and the more corrosion-resistant the material is [47,48]. Therefore, sample C showed better corrosion resistance than the bare 304 SS.

The potentiostatic polarization curves for sample C and the bare 304 SS in a simulated PEMFC environment are shown in Figure 11. Figure 11a shows the potentiostatic polarization of sample C and the bare 304 SS at a potential of 0.6 V in a simulated PEMFC cathode environment. The corrosion current density of sample C decreased slightly and fluctuated around $0.035\;\mu A\cdot {\mathrm{cm}}^{-2}$, whereas the corrosion current density of the bare 304 SS began to decrease sharply and then gradually stabilized at approximately $0.44\;\mu A\cdot {\mathrm{cm}}^{-2}$. Figure 11b shows the results of the potentiostatic polarization test for sample C and the bare 304 SS at a −0.1 V potential under a simulated PEMFC anode operation. The corrosion current density of sample C initially increased slightly and gradually stabilized at $-0.042\;\mu A\cdot {\mathrm{cm}}^{-2}$. This cathodic current indicates that sample C was cathodically protected and that the active dissolution rate in the anodic environment was very low. The corrosion current density of the bare 304 SS changed similarly to the corrosion current density in the cathodic environment by dropping sharply initially and then stabilizing at $1.3\;\mu A\cdot {\mathrm{cm}}^{-2}$. At the beginning of the polarization of the bare 304 SS, the current density decreased rapidly with time and finally stabilized. This is because the surface of the passivation film started to form when the bare 304 SS was initially polarized and the rate of the formation of the passivation film was less than the dissolution rate. As the polarization proceeded, the formation of the passivation film gradually increased, as the film formation rate was greater than the film dissolution rate. Ultimately, the passivation film completely covered the stainless steel surface, and the current required to maintain the passivation film was small. However, the current density of sample C was maintained at a low level because the surface of sample C was covered by vertical CNTs and the contact area of the etched solution was small. Also, the etching solution did not react with the CNTs. The experimental results for the corrosion resistance test of the sample also show that the corrosion resistance of sample C improved, and that the corrosion current density was lower than the target of $1\;\mu A\cdot {\mathrm{cm}}^{-2}$ proposed by the U.S. Department of Energy (DOE).

The initial performances of the cells with bipolar plates, with or without the application of vertical CNTs, are shown in Figure 12. The open circuit voltages of the two fuel cells were both at 0.913V. The output voltage of bare 304 SS attenuated faster than that of the 304 SS with vertical CNTs. The different voltages between them mainly occurred in the ohmic and concentration polarization regions. The voltage difference in the ohmic polarization region was caused by contact resistance, as shown in Figure 9. The metal ions from the bare 304 SS contaminated the proton exchange membrane, which weakened the chemical reaction and led to voltage loss in the concentration polarization region.

Figure 13 shows the performance of two cells during five cold-start cycles. No performance loss was observed when the current density was less than 200 mA·cm^−2^, and the voltage loss of bare 304 SS increased with the number of cold-start cycles, as shown in Figure 13a. After the third time, the cell could not start. The reason for this is that the metal ions ionized from bare 304 SS as they were polluted by the proton exchange membrane, resulting in a weakened cell reaction. Further, ice formed by water in the flow channel prevented the gas from entering the reaction region, which led to the sharp attenuation in the concentration polarization region. Figure 13b shows that the maximum voltage loss of the cell with the vertical CNT 304 SS bipolar plate was about 0.04V during five cold-starts. Due to the superhydrophobic property of vertical CNTs, the cell could reduce the formation of ice and quickly remove water at low temperatures. Moreover, the vertical CNTs had good corrosion resistance, which ensured that cell performance would not be irreversibly attenuated with the number of experiments.

## 4. Conclusions

In this paper, methane was used as a carbon precursor and different sizes of nanosized stainless steel particles on pretreated 304 SS were used as catalysts. CNTs with different morphologies were grown directly on the surface of 304 SS by PECVD. By contrast, the surface of the 304 SS with vertical CNTs had the best hydrophobicity out of all of the samples, and the contact angle reached 150°, which was greatly improved when compared with the contact angle of 83.5° for the bare 304 SS. The icing time for the 304 SS with vertical CNTs was the longest in the anti-icing experiment and reached 1638 s, which was nearly four times the freezing time of bare 304 SS. The results from the voltammetric comparison of the surface of 304 SS with vertical CNTs and the bare 304SS show that the interface contact resistance decreased by one order of magnitude. In the simulated PEMFC environment, the results show that the potentiostatic polarization shifted by 0.35 V and the corrosion current density reduced by 90%. These results indicate that the corrosion resistance of the modified sample was greatly improved. The method for directly growing vertical CNTs on the surface of stainless steel by PECVD greatly improved the hydrophobic properties, anti-icing ability, electrical conductivity and corrosion resistance. Using −10 °C and RH 30% reaction gas, the cell performance results revealed that the bare 304 SS bipolar plate showed irreversible attenuation, while the performance of the 304 SS bipolar plate with vertical CNTs was hardly affected. Therefore, the growth of vertical CNTs on 304 SS as a low-temperature PEMFCs bipolar plate has good prospects for future applications.

## Figures and Tables

**Figure 1 materials-12-00899-f001:**
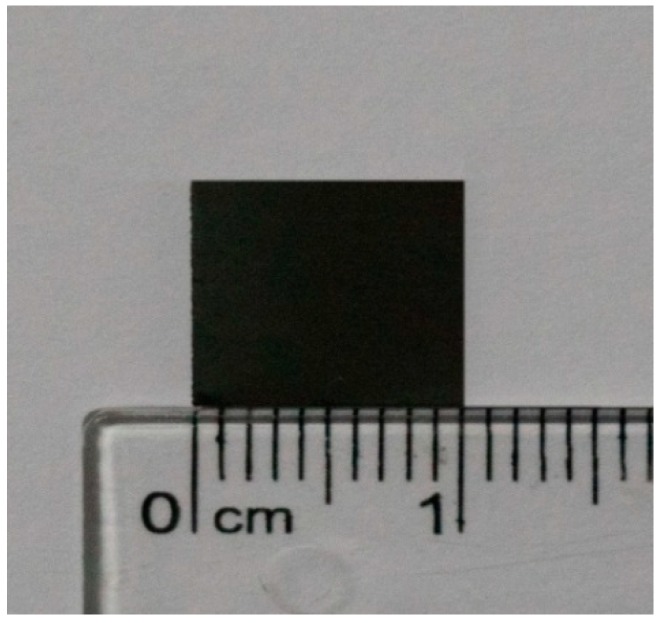
Stainless steel sample with CNTs.

**Figure 2 materials-12-00899-f002:**
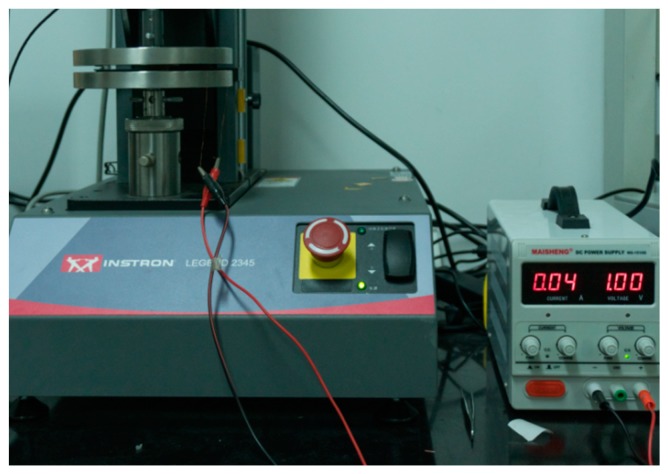
Contact resistance testing device.

**Figure 3 materials-12-00899-f003:**
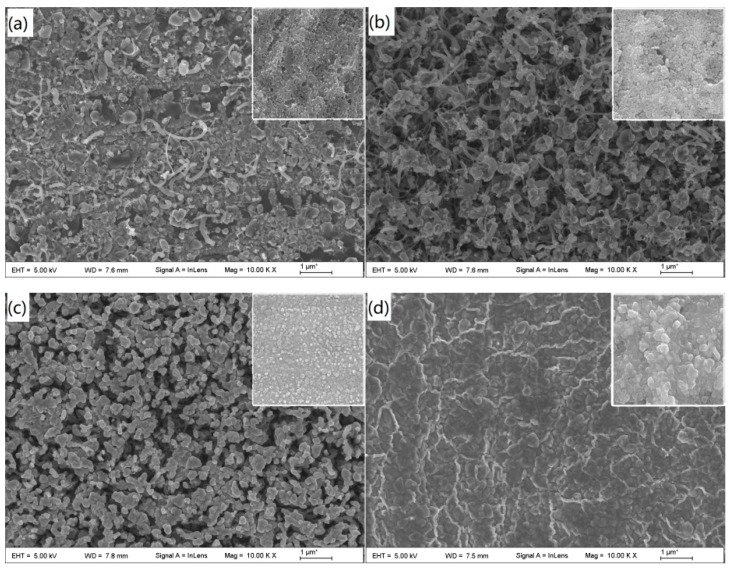
Carbon nanotube morphology on the surface of the sample at different methane ratios: (**a**) Sporadic CNTs grew on the surface of sample A; (**b**) a large number of wound CNTs grew on the surface of sample B; (**c**) vertical CNTs grew on the surface of sample C; (**d**) no CNT growth on the surface of sample D.

**Figure 4 materials-12-00899-f004:**
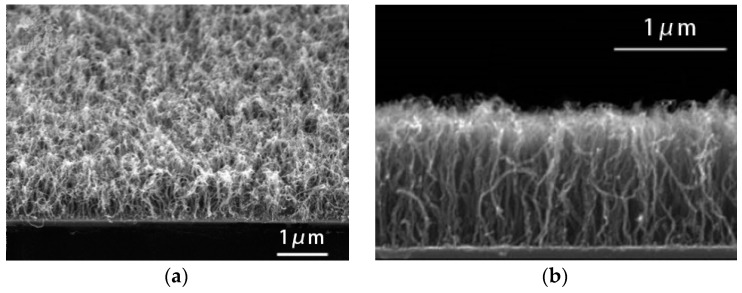
Cross-sectional SEM image of the grown carbon nanotube samples: (**a**) A large number of wound CNTs grew on the surface of sample B; (**b**) vertical CNTs grew on the surface of sample C.

**Figure 5 materials-12-00899-f005:**
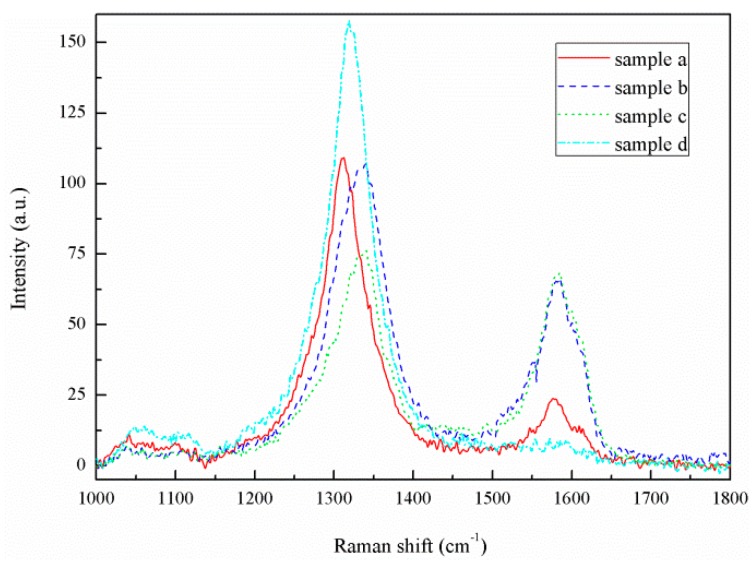
Raman spectra for the modified sample.

**Figure 6 materials-12-00899-f006:**
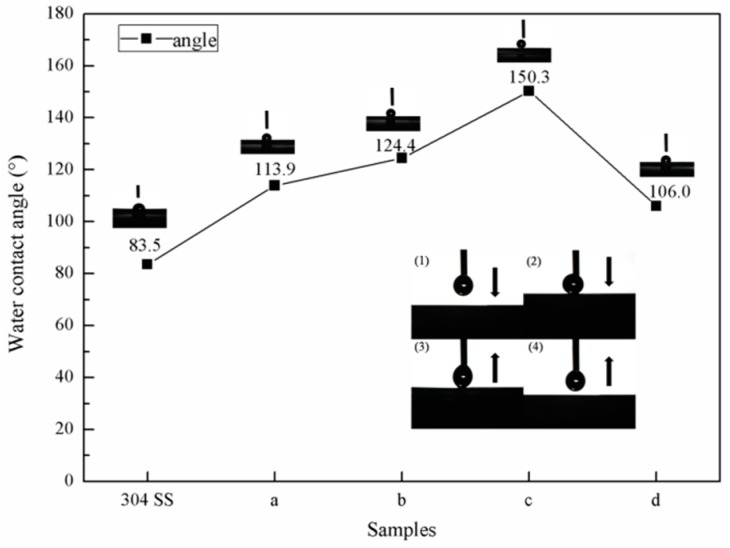
Static contact angle on the sample surface.

**Figure 7 materials-12-00899-f007:**
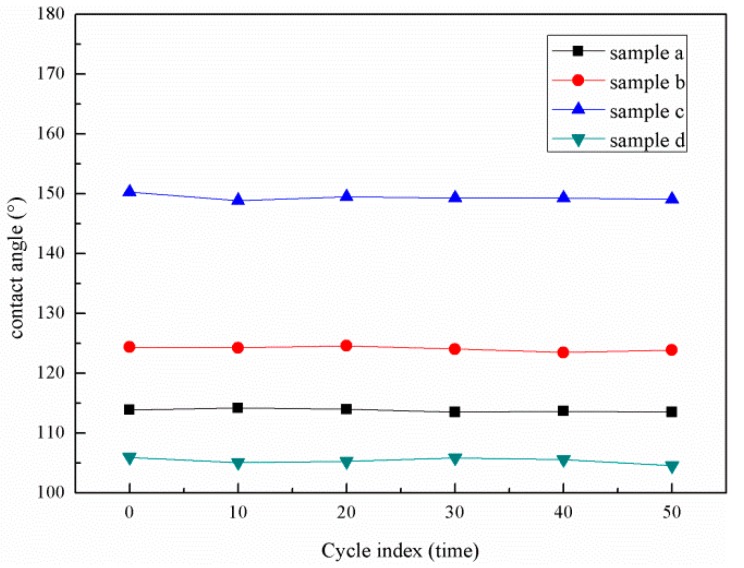
Effect of alternating high and low temperatures on the hydrophobicity of the samples.

**Figure 8 materials-12-00899-f008:**
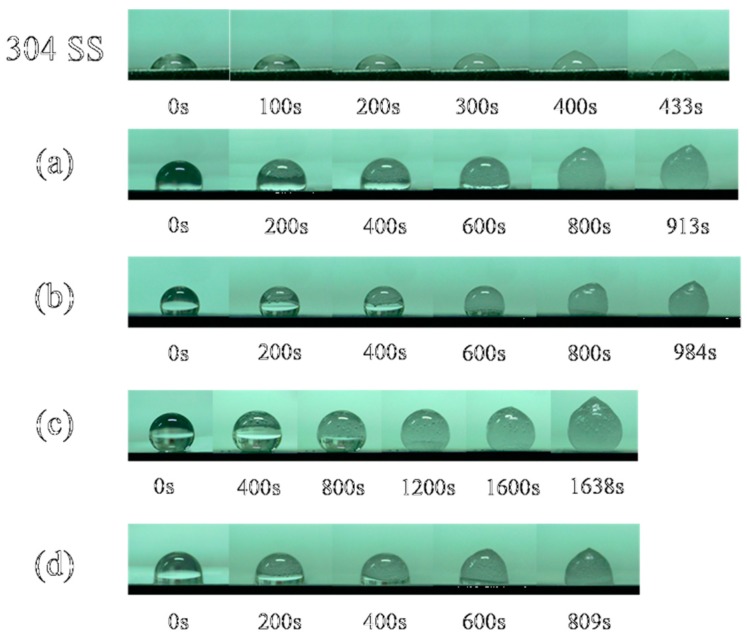
In situ icing on the surface of the sample.

**Figure 9 materials-12-00899-f009:**
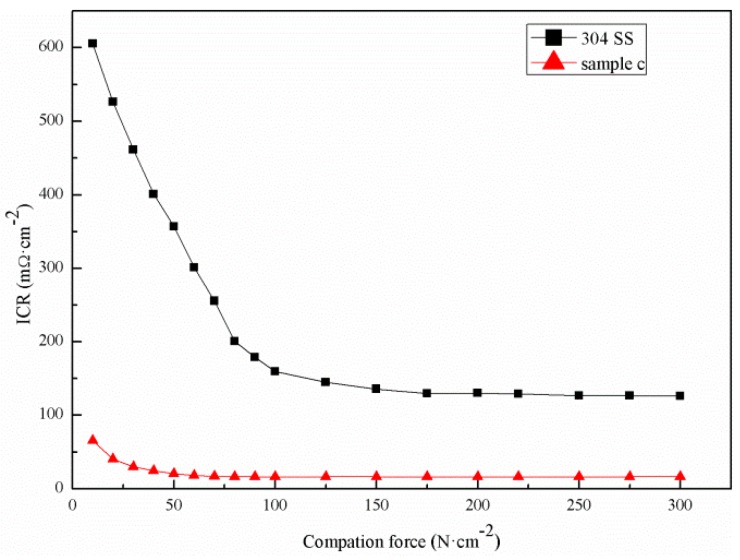
Contact resistance between the sample and the carbon paper at different pressures.

**Figure 10 materials-12-00899-f010:**
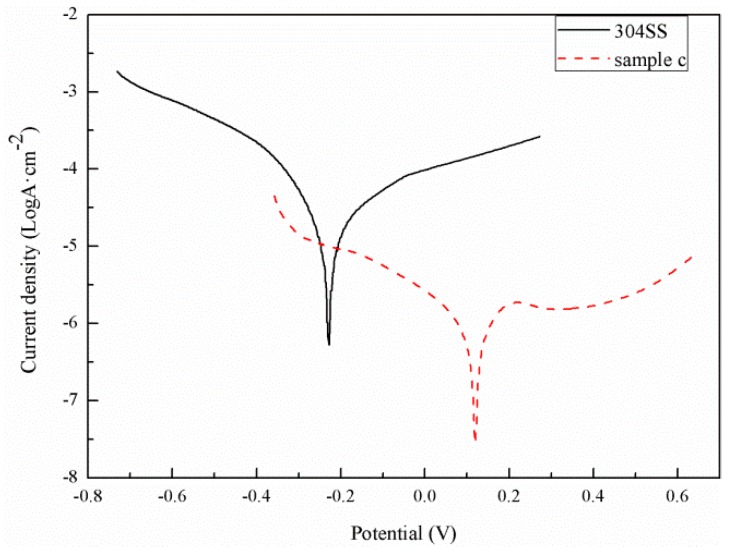
Potentiokinetic curves of the samples simulated in a proton exchange membrane fuel cell (PEMFC) environment.

**Figure 11 materials-12-00899-f011:**
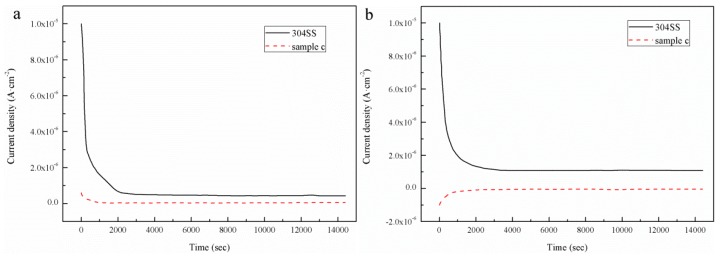
Potentiostatic polarization curves for sample C and the 304 stainless steel in a simulated PEMFC: (**a**) 0.6-V cathodic environment and (**b**) −0.1-V anodic environment.

**Figure 12 materials-12-00899-f012:**
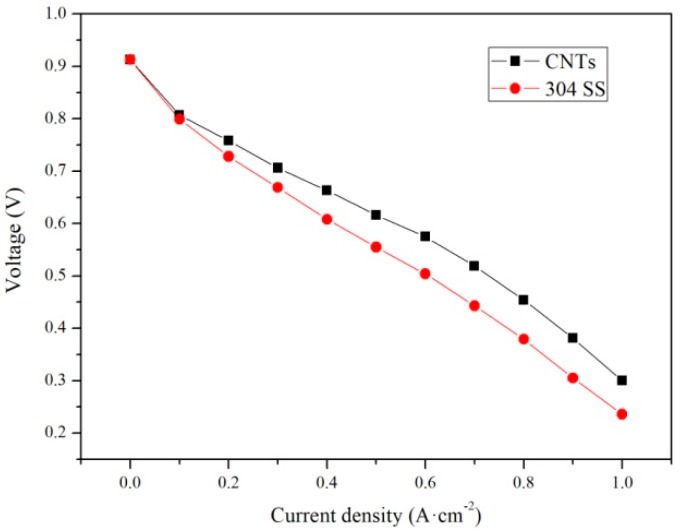
Initial performance of the two cells: Bare 304 SS as the bipolar plate; 304 SS with vertical CNTs as the bipolar plate.

**Figure 13 materials-12-00899-f013:**
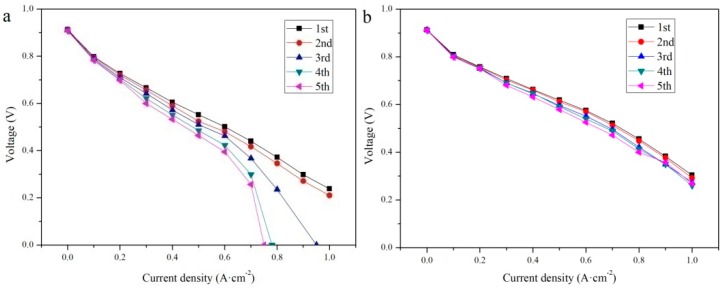
Performances of the cells: (**a**) Bare 304 SS as the bipolar plate and (**b**) 304 SS with vertical CNTs as the bipolar plate.

**Table 1 materials-12-00899-t001:** Growth parameters for the carbon nanotubes (CNTs) on stainless steel surfaces.

Samples	Plasma Etching Time/min	Growth Pressure/mT	Growth Time/min	$CH_{4}:H_{2}/\mathrm{sccm}$	Growth Temperature/°C
a	0	214–256	20	6:14	700
b	5	223–263	20	6:14	700
c	10	198–236	20	6:14	700
d	20	226–269	20	6:14	700

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
