# Peer review of "Study on the Properties of Vertical Carbon Nanotube Films Grown on Stainless Steel Bipolar Plates"

_materials, 2019, doi:10.3390/ma12060899_

Round 1
Reviewer 1 Report
Please improve the material: 1) give details of deposition (total pressure, temperature,geometry of reactor, substrate - HF superposition etc.) 2) show the cross-section SEM-images. 3) It's necessary to evident the creation/absence of CNT. Show the SEM with high-quality or TEM for single or 1-4 samples.
Author Response
Q1: give details of deposition (total pressure, temperature,geometry of reactor, substrate - HF superposition etc.)
Response1:We have studied reviewer’s comments carefully and made red marked revisions in lines 106, 109 of the paper.
Q2: show the cross-section SEM-images.
Response2:We have added SEM-images in Figure4.
Q3: It's necessary to evident the creation/absence of CNT
Response3:The creation/absence of CNT can be seen in Figure4.
Reviewer 2 Report
This manuscript demonstrates that vertical carbon nanotube (CNT) films grown on stainless steel bipolar plates enhance the hydrophobic and anti-icing properties, reduce the contact resistance, and improve the corrosion resistance. This would be effective at enhancing the performance of a proton exchange membrane fuel cell (PEMFC). However, the manuscript could be revised according to the following comments before being published.
(1) The authors had better indicate some fuel cell polarization curves obtained with the bipolar plates with and without the application of vertical CNTs.
(2) The authors demonstrate that bipolar plates with super-hydrophobic flow channels are effective at preventing the water droplets from freezing under low-temperature starting conditions. However, it is also essential to enhance the water management properties and prevent flooding under normal operating temperature conditions. There are some published papers which demonstrate that bipolar plates with super-hydrophilic flow channels are effective at promoting the discharge of excess water in the gas diffusion layers and the flow channels, thereby enhancing the cell performance. Therefore, the authors had better introduce the previous studies on the hydrophilic flow channels in the introduction section.
Author Response
Q1: The authors had better indicate some fuel cell polarization curves obtained with the bipolar plates with and without the application of vertical CNTs.
Response 1:We have studied reviewer’s comments carefully and added content on the polarization curve is located in lines 190 and 380 of the text.
Q2:The authors demonstrate that bipolar plates with super-hydrophobic flow channels are effective at preventing the water droplets from freezing under low-temperature starting conditions. However, it is also essential to enhance the water management properties and prevent flooding under normal operating temperature conditions. There are some published papers which demonstrate that bipolar plates with super-hydrophilic flow channels are effective at promoting the discharge of excess water in the gas diffusion layers and the flow channels, thereby enhancing the cell performance. Therefore, the authors had better introduce the previous studies on the hydrophilic flow channels in the introduction section.
Response 2:If I am not mistaken, the research content on the introduction of hydrophobic flow channels is increased in line 60 of the text.
Reviewer 3 Report
The work by Lu and coworkers is well-motivated, and its significance is clearly communicated. The text is largely easy to follow and is organized in a logical way. Some copy-editing is needed but not much. The experiment appears to be done well, and the conclusions drawn are straightforward based on the presented data. My one criticism is that the Conclusions need to have a more direct and quantitative statement about the types of conditions (temperature, humidity levels, etc.) where vertical CNTs on 304 SS would work well. This is hinted in the Abstract (portable power supplied in Northern China), and it leaves the reader wondering these materials would work in this and related environments.
Author Response
Q:Conclusions need to have a more direct and quantitative statement about the types of conditions (temperature, humidity levels, etc.) where vertical CNTs on 304 SS would work well.
Response: We have studied reviewer’s comments carefully and made a direct and quantitative statement of the types of conditions in which the cell operates in the conclusion. We have tried our best to revise our manuscript according to the comments. Attached please find the revised version, which we would like to submit for your kind consideration.